# Peptide Nucleic Acid (PNA)-Enhanced Specificity of a Dual-Target Real-Time Quantitative Polymerase Chain Reaction (RT-qPCR) Assay for the Detection and Differentiation of SARS-CoV-2 from Related Viruses

**DOI:** 10.3390/diagnostics10100775

**Published:** 2020-09-30

**Authors:** Won-Suk Choi, Ju Hwan Jeong, Halcyon Dawn G. Nicolas, Sol Oh, Khristine Joy C. Antigua, Ji-Hyun Park, Beomkyu Kim, Sun-Woo Yoon, Kyeong Seob Shin, Young Ki Choi, Yun Hee Baek, Min-Suk Song

**Affiliations:** 1Department of Microbiology, Chungbuk National University College of Medicine and Medical Research Institute, Cheongju 28644, Korea; tuckgirlee@naver.com (W.-S.C.); jeongbau07@naver.com (J.H.J.); hdnicolasdvm@gmail.com (H.D.G.N.); 10356ok@naver.com (S.O.); tineantigua@gmail.com (K.J.C.A.); jihyun8943@naver.com (J.-H.P.); rick960426@naver.com (B.K.); choiki55@cbnu.ac.kr (Y.K.C.); 2Infectious Diseases Research Center, Korea Research Institute of Bioscience and Biotechnology, Daejeon 34141, Korea; syoon@kribb.re.kr; 3Department of Laboratory Medicine, Chungbuk National University College of Medicine, Cheongju 28644, Korea; ksshin@chungbuk.ac.kr

**Keywords:** SARS-CoV-2, SARS related-CoV, PNA, RT-qPCR, dual-target detection

## Abstract

The threat posed by coronaviruses to human health has necessitated the development of a highly specific and sensitive viral detection method that could differentiate between the currently circulating severe acute respiratory syndrome coronavirus 2 (SARS-CoV-2) and other SARS-related coronaviruses (SARSr-CoVs). In this study, we developed a peptide nucleic acid (PNA)-based real-time quantitative polymerase chain reaction (RT-qPCR) assay targeting the N gene to efficiently discriminate SARS-CoV-2 from other SARSr-CoVs in human clinical samples. Without compromising the sensitivity, this method significantly enhanced the specificity of SARS-CoV-2 detection by 100-fold as compared to conventional RT-qPCR. In addition, we designed an RT-qPCR method for the sensitive and universal detection of ORF3ab-E genes of SARSr-CoV with a limit of detection (LOD) of 3.3 RNA copies per microliter. Thus, the developed assay serves as a confirmative dual-target detection method. Our PNA-mediated dual-target RT-qPCR assay can detect clinical SARS-CoV-2 samples in the range of 18.10–35.19 Ct values with an 82.6–100% detection rate. Furthermore, our assay showed no cross-reactions with other coronaviruses such as human coronaviruses (229E, NL63, and OC43) and Middle East respiratory syndrome coronavirus, influenza viruses (Type B, H1N1, H3N2, HPAI H5Nx, and H7N9), and other respiratory disease-causing viruses (MPV, RSV A, RSV B, PIV, AdV, and HRV). We, thus, developed a PNA-based RT-qPCR assay that differentiates emerging pathogens such as SARS-CoV-2 from closely related viruses such as SARSr-CoV and allows diagnosis of infections related to already identified or new coronavirus strains.

## 1. Introduction

Betacoronavirus belongs to the family Coronaviridae and is distinguished by its large, non-segmented, positive single-stranded RNA, which is approximately 26–30 kb long [1,2]. Betacoronaviruses were thought to mainly infect animals and cause occasional mild infections in humans [1,3] until the outbreaks of severe acute respiratory syndrome coronavirus (SARS-CoV) and Middle East respiratory syndrome coronavirus (MERS-CoV), two highly infectious zoonotic diseases that caused major epidemics worldwide [3,4]. These viruses, along with other genetically diverse SARS-related coronaviruses (SARSr-CoV) and MERS-related coronaviruses (MERSr-CoV), have been transmitted from their natural hosts such as bats and rodents to an intermediate animal host of different species such as civet cats and dromedary camels before being transmitted to a human host [3,4,5,6,7,8,9]. Genetically diverse coronaviruses have been known to cause infections not only in humans but also in animals [10,11]. Given the predisposition of coronaviruses, particularly betacoronaviruses, to switch hosts [5], an emergence or re-emergence of coronavirus disease or the possibility of another pandemic is highly plausible. 

In December 2019, SARS-CoV-2, classified by the Coronaviridae Study Group of the International Committee on Taxonomy of Viruses as a SARSr-CoV belonging to the subgenus *Sarbecovirus* [12,13,14], sparked an outbreak of COVID-19. Initially known as viral pneumonia, COVID-19 subsequently spread from Wuhan, Hubei Province of China, infecting more than 13 million people from over 213 countries. Since then, COVID-19 is a pandemic with a fatality rate of more than 4% [15]. 

The emergence of the novel SARS-CoV-2 with its efficient human-to-human transmission and ability to cause asymptomatic infections [5,14,16,17,18] has highlighted the need to develop a highly specific and sensitive viral detection method to mitigate the potential contact with infected individuals and prevent the further spread of infection. At present, real-time quantitative polymerase chain reaction (RT-qPCR) is widely used to detect SARS-CoV-2 and plays a critical role in viral containment [13,19,20,21]. Other nucleic acid amplification methods such as reverse transcription loop-mediated isothermal amplification (RT-LAMP) [22,23] have been used as supportive molecular diagnostic methods. Although conventional RT-qPCR is the gold standard for the detection of infectious agents such as SARS-CoV-2 [24,25,26,27], confirmatory diagnosis may be challenging as it may not efficiently discriminate SARS-CoV-2 from SARS-CoV and other SARSr-CoVs because of the viruses’ close genetic relatedness, reported to be 82% and 89%, respectively, [28]. Moreover, the currently limited information on the genetic diversity of coronaviruses infecting humans and animals, and the tendency of coronaviruses to jump hosts and cause a pandemic, as exhibited by SARS-CoV-2 today, press the need to develop a viral detection method that can mitigate the cross-reactive detection with the currently circulating SARS-CoV-2 and the potent SARSr-CoV.

Peptide nucleic acid (PNA) is a synthetic analog where the normal phosphodiester backbone is replaced by a 2-aminpethylglycine chain and its nucleobases complement DNA or RNA by the normal AT and GC geometry. As PNA cannot serve as a primer for polymerization and exhibits higher affinity (up to 1000-fold) for target complementary DNA or RNA sequences than primers and probes [29], it may block annealing or chain elongation in PCR [30,31,32]. In the present study, we developed a PNA-mediated RT-qPCR assay to efficiently discriminate the SARS-CoV-2 N gene from other SARSr-CoV genes. We exploited the blocking effect of PNA in PCR to design a primer set targeting a specific N gene sequence with improved specificity through the inhibition of SARS-CoV and SARSr-CoV amplification, without any compromise in the sensitivity to detect SARS-CoV-2. 

To our knowledge, this is the first attempt to use PNA-mediated RT-qPCR to improve the specificity of SARS-CoV-2 detection. In addition, we also improved the universal RT-qPCR assay targeting the more conserved regions from the ORF3ab gene up to the E gene of SARSr-CoV viruses to span all the known strains of the subgenus *Sarbecovirus*.

## 2. Materials and Methods 

### 2.1. Primers, Probes, and PNA Blocker Design for Dual-Target Detection

The dual-target primers and probe were designed based on the highly conserved sites of the aligned sequences. In particular, the primers, probes, and PNA blocker were designed based on ORF3ab and E gene sequences of SARS-CoV/HKU-39849 (SARS-CoV; 25943–26333 bp), BetaCoV/Wuhan-Hu-1/2019 (SARS-CoV-2; 26142–26532 bp), and Beta-CoV/Bat/HKG/HKU3_8/2010 (Bat/SARSr-CoV; 22970–26268 bp) and the N protein sequence of SARS-CoV (28033–28430 bp) and SARS-CoV-2 (28224–28623 bp). In addition, both ends of the probe were modified with 6-carboxyfluorescein and Black Hole Quencher. GC ratio and melting temperature values were considered during the design before custom synthesis (Bionics Inc., Daejeon, Korea). For the PNA blocker design, SARSr-CoV N blocker was designed based on the sequences of SARSr-CoV (100% consensus) and SARS-COV-2 (82.3% consensus) and then custom synthesized by Panagene Inc. (Daejeon, Korea).

### 2.2. Viral Propagation in Cell Lines and RNA Retrieval

Beta-CoV/Korea/KCDC03/2020, Beta-CoV/Korea/NMC01/2020, and Beta-CoV/Korea/NMC02/2020 viruses were propagated in Vero cells in Dulbecco’s modified Eagle’s medium (Gibco-Invitrogen, Carlsbad, CA, USA) supplemented with 10% fetal bovine serum and 1% antibiotics, and maintained at 37 °C. For synthetic SARS-CoV-2 DNA, PCR amplification using primers with extended T7 promoter was performed as previously reported [22]. In brief, RNA transcription of the amplified synthetic DNA was performed using T7-MEGAscript^®^ kit (Invitrogen, Vilnius, Lithuania), and the RNA copy number was measured using Qubit™ RNA-HS Assay Kit (Thermo Fisher Scientific Inc., Waltham, MA, USA) as directed. Viral RNA was extracted from the infected cells and clinical samples using the RNeasy Kit (QIAGEN, Hilden, Germany) according to the manufacturer’s instructions.

### 2.3. SARS-CoV-2 Viruses and Clinical Specimens

Nasal swab samples were obtained from SARS-CoV-2-confirmed patients confined at the Chungbuk National University Hospital. To test the cross-reactivity of our designed primer sets with other respiratory disease-causing viruses, confirmed clinical samples were used to establish a panel comprising human coronavirus (229E, NL63, and OC43), MERS-CoV, influenza A/California/04/2009 (H1N1 pdm), A/Perth/16/2009 (H3N2), B/Brisbane/60/2008 (Victoria Lineage), B/Phuket/3073/2013 (Yamagata Lineage), highly pathogenic avian influenza viruses (H5N1, H5N6, H5N8, and H7N9), adenovirus (AdV), parainfluenza virus (PIV), human metapneumovirus (MPV), human rhinovirus (HRV), and respiratory syncytial virus A and B (RSV A and B).

### 2.4. Optimization of Primer and Probe Concentrations

For RT-qPCR, we selected the dual-target detection primers and probes with the highest sensitivity among the primer–PNA combinations of varying concentrations and with different annealing temperatures. Thus, four sets of ORF3ab-E detection primers and two sets of N detection primers at concentrations from 0.1 to 0.5 μM were used to amplify the templates of the synthesized RNAs from SARS-CoV, SARS-CoV-2, and Bat-SARSr-CoV ORF3ab-E and N genes and the viral RNAs from SARS-CoV and SARS-CoV-2 at 1 × 10^5^ dilution. False-positive experiments were repeated thrice using RNase-free water as negative control. The primer set without any false-positive detection for each target was then selected as the final primer set.

### 2.5. RT-qPCR Condition Optimization

As the PNA annealing temperature could affect the performance of RT-qPCR and produce false-positive results, the optimization of the PNA annealing temperature is warranted. To determine the most optimal annealing temperature for PNA, PCR reactions were carried out with and without the PNA blocker using a temperature gradient under the following conditions: 30 min at 50 °C for reverse transcription, 15 min at 95 °C for PCR initial reaction, followed by 40 cycles of 10 s at 95 °C for denaturation, 10 s at 66 °C to 76 °C for PNA binding, 10 s at 50 °C for primer binding, and 10 s at 72 °C for elongation. The final primer, probe, and PNA concentrations used in the optimization experiments are shown in Table 1, while PCR conditions are shown in Figure 1B.

### 2.6. Sensitivity and Specificity Confirmation of the PNA-Mediated Dual-Target RT-qPCR Assay

All RT-qPCR assays were performed on the CFX96 Real-Time PCR Detection System (Bio-Rad^®^ Laboratories, Inc., Hercules, CA, USA) using the TOPreal™ One-step RT-qPCR kit (Enzynomics Co., Daejeon, Korea). Each 20 μL reaction mixture comprised 5 μL TOPreal™ One-step RT-qPCR kit reagent, 1 μL forward primer (250 nM), 1 μL reverse primer (250 nM), 1 μL probe (250 nM), and 1 μL of SARSr-CoV PNA blocker for N gene detection (500 nM), up to 20 μL of RNase-free water, and 3 μL RNA template. The thermal cycling condition was 30 min at 50 °C for reverse transcription, 15 min at 95 °C for PCR initial reaction, followed by 40 cycles of 10 s at 95 °C for denaturation, 10 s at 73.4 °C for PNA binding, 10 s at 50 °C for primer binding, and 10 s at 72 °C for elongation.

To confirm the detection sensitivity, the synthetic RNAs of SARS-CoV, SARS-CoV-2, and Bat/SARSr-CoV were serially diluted from 1 × 10^8^ to 1 × 10^0^ RNA copies per 3 μL per reaction, while SARS-CoV-2 RNA was serially diluted from 1 × 10^−8^ to 1 × 10^−1^. The serially diluted RNAs were then used as templates for the PNA-mediated dual-target RT-qPCR. Each reaction was performed in triplicate.

To determine the detection specificity, dual-target RT-qPCR was performed using the RNAs of the abovementioned strains in addition to the RNAs from three human coronaviruses, eight influenza viruses, and six other respiratory disease-causing viruses. Each viral RNA was confirmed through RT-qPCR primer sets capable of detecting these viruses. Furthermore, we used SARS-CoV-2 viral RNA as control for the NIID-Japan RT-qPCR detection method [33] (Appendix A).

### 2.7. Ethics Statement

Clinical samples were collected from the National Medical Center, South Korea, under the approved guidelines and relevant regulations. All experiments were performed following the approved guidelines from the Ethics Committee and Institutional Review Board of National Medical Center (IRB no. H-2002-111-002, approved on 5 February 2020).

## 3. Results

### 3.1. Primer, Probe, and PNA Blocker Design

To design the primers and probes, the sequences of SARS-CoV-2 (n = 594) and SARSr-CoV (n = 332, except laboratory strains) of different origin, including human, bat, civet, monkey, badger, pangolin, and anteater, were downloaded from the GenBank and Global Initiative on Sharing All Influenza Data (GISAID) databases. We primarily attempted to design specific primers and a probe set targeting the SARS-CoV-2 N gene to differentiate SARS-CoV-2 from SARSr-CoV. Thus, the highly conserved regions from the SARS-CoV-2 sequences that were different from other SARSr-CoV sequences were analyzed. In silico analysis of the aligned sequences showed that the N gene is highly conserved among SARS-CoV-2 viruses. However, the actual alignment revealed the variations and mismatches between SARS-CoV-2 and SARSr-CoV N genes, with a computed primer-binding region sequence similarity of 76.4–81.8% (Figure 1A). To improve the specificity of the RT-qPCR method for SARS-CoV-2, a 17-nucleotide long PNA blocker located at the conserved region between the forward primer and probe of the SARSr-CoV N gene was additionally designed based on the aligned SARSr-CoV sequences other than the SARS-CoV-2 sequence (Figure 1B). This approach would improve the detection specificity for the SARS-CoV-2 N gene because of the arrested amplification of SARSr-CoV-2 N genes by the PNA blocker.

To develop our detection assay as a dual-targeting confirmatory diagnostic method, we additionally designed RT-qPCR primers and a probe set that targeted genes within the highly conserved regions of SARSr-CoV between ORF3ab and E genes. The alignment of 20 representative sequences of SARS-CoV-2 and SARSr-CoV belonging to the subgenus *Sarbecovirus* demonstrated high conservation in the binding regions of the designed universal primers (Figure 2a). Hence, the in silico cross-reactivity would be highly efficient based on the sequence similarity of the primer-binding regions bearing zero mismatches among 332 strains (100%) for almost all the designed primer and probe sets except for the reverse primer with one strain (Beta-CoV/Bat/HKG/HKU3_8/2010), which was observed with one nucleotide difference. Moreover, as shown in Figure 2b, a phylogenetic tree was constructed based on the selected representative sequences to illustrate the detection coverage of our designed universal ORF3ab-E gene primer and probe set.

To optimize the primers designed for the PNA-mediated RT-qPCR assay, 10-fold serial dilutions of SARS-CoV and SARSr-CoV-2 RNA transcripts (1 × 10^0^ to 1 × 10^8^ RNA copies per 3 μL per reaction) were used as templates. The primers and probe designed for SARS-CoV-2 could detect 10 RNA copies per reaction. (Figure 3a), while those specific for SARS-CoV had a detection limit of 10^6^ RNA copies per reaction (Figure 3b). The amplification curves showed that the designed primer pairs and probe could effectively detect the N gene of SARS-CoV despite the 18.2–23.6% sequence mismatches in the primer design (Figure 3).

To assess the capability of the designed PNA blocker to discriminate the SARS-CoV-2 N gene from the SARS-CoV N gene, we compared detection efficiency using 10-fold serially diluted SARS-CoV-2 and SARS-CoV RNA transcripts. In the absence or presence of the PNA blocker, the detection limit for SARS-CoV-2 remained unchanged and was 10 RNA copies per reaction (Figure 3a,c); however, the SARS-CoV detection was blocked, thereby requiring 100-fold higher RNA copies per reaction, i.e., from 10^6^ to 10^8^ RNA copy number per reaction (Figure 3b,d). As hypothesized, the detection of the SARS-CoV N gene was efficiently inhibited by the PNA blocker. 

To determine the limit of detection (LOD) for the SARS-CoV-2 primer and probe set, 20 repetitions of reactions with 100, 20, 10, and 1 RNA copies per 3 μL of the SARS-CoV-2 N gene transcript with or without the PNA blocker were tested. We found that the detection rate was 100% with 100 and 20 RNA copies per reaction, 80–90% with 10 RNA copies per reaction, and 0% with 1 RNA copy per reaction, indicating a relative LOD of 6.7 RNA copies per μL (Figure 3e). Furthermore, we also determined the effectiveness of the designed PNA blocker using high RNA copy numbers (1 × 10^9^ and 1 × 10^10^) of the SARS-CoV N gene transcript and observed the delayed detection of the SARS-CoV N gene in the presence of the PNA blocker (Appendix A). Taken together, the results present significant improvement in the detection specificity of the RT-qPCR assay in differentiating SARS-CoV-2 from SARSr-CoV using the PNA blocker.

### 3.2. Optimization and Sensitivity Evaluation of the Universal RT-qPCR Assay for ORF3ab-E Gene Detection 

To test the sensitivity of the designed primer and probe set for the universal detection and amplification of ORF3ab-E genes, the regions between the ORF3ab and E genes of serially (1 × 10^0^ to 1 × 10^8^ per 3 μL per reaction) diluted SARS-CoV-2, SARS-CoV, and Bat/SARSr-CoV RNA transcripts were amplified. The amplification curves of the SARS-CoV-2 ORF3ab and E genes demonstrated the ability of the designed universal detection primers and probe to readily detect 1 × 10^8^ RNA copies per reaction within 12 cycles. Nonetheless, in this study, the designed universal primer–probe set was highly sensitive to detect 10 RNA copies per reaction for SARS-CoV-2, SARS-CoV, and Bat/SARSr-CoV (Figure 4a–c).

To determine the LOD, 20 repetitions of 100, 10, and 1 RNA copies per reaction of the SARS-CoV-2 ORF3ab-E genes were performed and a 100% positive result for 100 and 10 RNA copies per reaction and a 0% positive result for 1 RNA copy per reaction were obtained. Thus, the LOD of the designed ORF3ab-E primers and probe was 3.3 RNA copies per μL (Figure 4d). The sensitivity and LOD of the universal primer set for SARS-CoV-2 detection were comparable with the sensitivity and LOD of the ORF3ab-E gene primer set for SARS-CoV and Bat/SARSr-CoV detection. This observation suggests that the designed universal detection primers and probes were highly sensitive in detecting the target viral gene at a detection limit of 3.3 RNA copies per μL. Further, the detection sensitivity was tested using a SARSr-CoV strain available in GenBank that was different by one nucleotide (Appendix A). The universal primer set could detect up to 100 RNA copies per reaction, indicating a 10-fold reduction in the detection sensitivity as compared to that observed with 100% matching strain sequences (Appendix A). In addition, the linearity was within the range of 10^1^ to 10^8^ RNA copies per reaction for SARS-CoV-2, SARS-CoV, and Bat/SARSr-CoV (Figure 4). Thus, our universal RT-qPCR method for the detection of SARSr-CoVs is not only highly sensitive but also extensive.

### 3.3. Performance Evaluation of the Dual-Target RT-qPCR Assay Using Intact Virus

The sensitivity and specificity of our dual-target RT-qPCR assay was further evaluated using intact viruses cultivated in vitro. The RNA was extracted from three SARS-CoV-2 cell culture samples, and then 10-fold (1 × 10^−2^ to 1 × 10^−8^) serially diluted. The sensitivity of the dual-target RT-qPCR in detecting the N and ORF3ab-E genes was analyzed in triplicates. Using the serially diluted viral RNAs, the sensitivity of the optimized N gene primers and probe was assessed. The PNA-mediated RT-qPCR assay targeting the N gene successfully detected all viruses at a minimum of 1 × 10^−7^ dilution, requiring a mean cycle threshold (Ct) value of 33.34 ± 0.43 to 33.41 ± 0.30 for each virus (Figure 5a). The sensitivity of the optimized ORF3ab-E gene primers and probe was also assessed, and mean Ct values from 32.10 ± 0.43 to 32.68 ± 1.06 at a minimum of 1 × 10^−7^ virus dilution were reported (Figure 5b).

We processed 20 repetitions of serially diluted intact viruses to validate the LOD of the designed primers for N and ORF3ab-E genes and performed RT-qPCR. In brief, the N and ORF3ab-E gene detection assays showed a positive detection rate of 100%, 90–100%, and 0–20% for 10^−6^, 10^−7^, and 10^−8^ dilution, respectively (Figure 5c,d). We compared the sensitivity of our dual-target RT-qPCR method with the sensitivity of the method suggested by the World Health Organization (WHO) [33] and observed comparable sensitivity for the three intact viral RNAs tested (Appendix A). 

### 3.4. Evaluation of the Dual-Target RT-qPCR Assay Using Clinical Samples

The accuracy of the optimized dual-target RT-qPCR primer pairs, probe, and PNA blocker was evaluated using clinical samples positive for SARS-CoV-2. Clinical samples positive for other respiratory disease-causing viruses, including representative coronaviruses, influenza viruses, and other respiratory viruses which include human metapneumovirus (MPV), respiratory syncytial virus (RSV) A and B, parainfluenza virus (PIV), adenovirus (AdV), and human rhinovirus (HRV), were also added to the panel. A total of 308 clinical and spike samples with confirmed hospital diagnosis were tested. The developed RT-qPCR assay clearly showed no cross-reaction with other respiratory disease-causing viruses, including major avian influenza viruses such as HPAI H5Nx and H7N9 (Table 2). The developed assay also proved to be effective in eliminating closely related coronaviruses such as MERS-CoV. All 23 clinical SARS-CoV-2 samples were detected at a range of 18.17 to 37.93 Ct values with a detection rate of 100% and 82.6% for universal ORF3ab-E and N genes, respectively (Table 2). These 23 hospital confirmed clinical samples were also cross-checked using the National Institute of Infectious Diseases (NIID) Japan RT-qPCR assay method. From the cross validation, results showed that 17 out of 23 (73.9%) clinical samples were detected positive for SARS-CoV-2, with a Ct value range of 18.10–36.40 (Appendix A).

## 4. Discussion

Molecular methods such as RT-qPCR are the gold standard for the detection of SARS-CoV-2, as declared by the WHO. These methods are specifically designed to target previously recognized coronaviruses but may have limitations in detecting novel and emerging coronaviruses [35]. Our newly developed PNA-mediated dual-target RT-qPCR assay addresses the need for an accurate diagnostic method for large-scale testing combined with the ability to distinguish true-positive SARS-CoV-2 infection from suspected cases with clinical presentations similar to SARSr-CoV, influenza, and other respiratory disease-causing viruses.

The incorporation of PNA as a blocker allowed us to develop an RT-qPCR assay with 100-fold enhanced specificity for SARS-CoV-2 detection which cross-reacted with more than 10^8^ RNA copy numbers of SARS-CoV (Figure 3). This amount of viral RNA would not be normally available in clinical samples as reported [36,37,38,39], suggesting that our PNA-mediated RT-qPCR will hardly demonstrate false positives from clinical samples of other SARSr-CoV. Our results prove that the PNA blocker clearly enhanced the detection of the SARS-CoV-2-specific N gene with a calculated sensitivity of 82.6%, a positive predictive value (PPV) of 100%, and a negative predictive value (NPV) of 98.7%. In addition, we ran the same set of clinical samples using the NIID-Japan RT-qPCR, which is widely reported as one of the sensitive methods for the detection of SARS-CoV-2 [34], and calculated a 74% detection rate which is comparable to our PNA-mediated RT-qPCR, suggesting its high sensitivity (Appendix A).

We were also able to demonstrate a notable improvement in the detection rate of SARS-CoV-2 with as low as 3.3 viral RNA copies per μL, which could be reliably detected (LOD). In comparison with the previously reported assay [40], our universal RT-qPCR assay was designed with the minimum primer mismatches using a more highly conserved region between the ORF3ab and E genes. Even a single nucleotide mismatch in the primer-binding region would result in a 10-fold decrease in the LOD. Thus, our primers retained the high sensitivity (LOD: 1 × 10^2^ RNA copies per 3 μL) for SARSr-CoVs and allowed us to develop an assay with high sensitivity for the detection of *Sarbecovirus* ORF3ab-E genes with 100% sensitivity, 100% PPV, and 100% NPV. We further evaluated our assay using synthesized RNAs of Bat/SARSr-CoVs and found improvement in the sensitivity for the detection of not just SARS-CoV-2 but also other related viruses within the subgenus *Sarbecovirus*. A limitation of this study is the lack of intact viral RNA for SARSr-CoV testing. Thus, future studies can explore the clinical applications of our assay. 

Overall, the PNA blocker greatly enhances the detection specificity of conventional molecular methods such as RT-qPCR. Our developed PNA-mediated dual-target RT-qPCR assay is a simple and highly sensitive method that is more accurate for SARS-CoV-2 diagnosis. The application of PNA in molecular detection methods such as RT-qPCR can significantly contribute to improving the ability to differentiate novel pathogens that are genetically close to each other, such as SARS-CoV-2 and other potentially related SARS-CoVs, as demonstrated in our study. Further, the detection of related CoVs, including those previously identified or yet undiscovered, may be possible. 

## Figures and Tables

**Figure 1 diagnostics-10-00775-f001:**
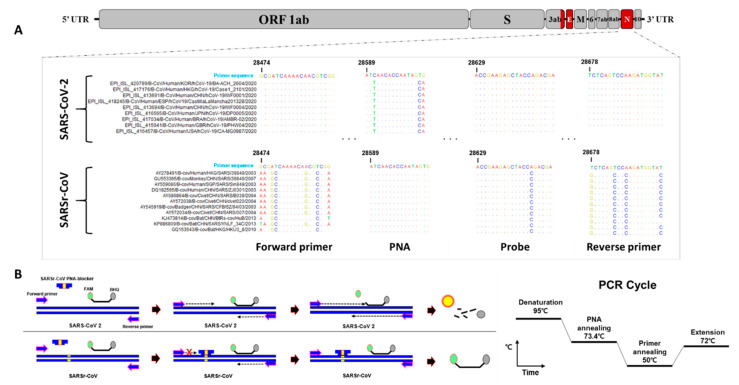
Correspondence between SARSr-CoV N gene sequences, highlighting the designed primer pair, probe, and blocker-binding site and the design principle. (**A**) Sequences of SARS-CoV-2 and SARSr-CoV were downloaded and aligned to identify the highly conserved regions of the N gene. The locations of the target regions of the N gene detection primer pairs, probe, and blocker from the target sequence positions: 28,474 (forward primer), 28,678 (reverse primer), 28,629 (probe), and 28,589 (blocker). Matching residues are represented as dots. (**B**) Peptide nucleic acid (PNA) blocker principle and RT-qPCR conditions. FAM: 6-carboxyfluorescein; BHQ: Black Hole Quencher^®^.

**Figure 2 diagnostics-10-00775-f002:**
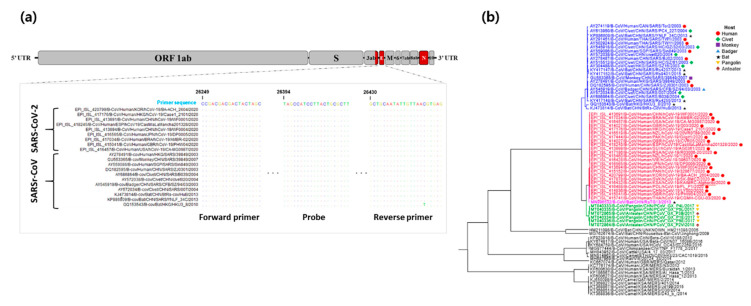
Correspondence between SARSr-CoV ORF3ab-E gene sequences, highlighting the designed primer pair and probe-binding site. (**a**) Representative sequences of SARS-CoV-2 and SARSr-CoV were aligned to identify the highly conserved regions of the ORF3ab-E gene. The target regions of the designed primer pair and probe were highlighted. Matching residues are represented as dots. (**b**) Sequence tree reconstructed using the BEAST program [34] to visualize the detection scope of the designed universal ORF3ab-E gene detection primers. The colored sequences correspond to SARSr-CoV constituents that can be detected by the developed assay. The hosts of each constituent strain are shown using a circle (human host), diamond (civet), square (monkey), triangle (badger), star (bat), inverted triangle (pangolin), and spade (anteater). 2.2 Optimization and sensitivity evaluation of the PNA-mediated RT-qPCR assay for N gene-specific detection.

**Figure 3 diagnostics-10-00775-f003:**
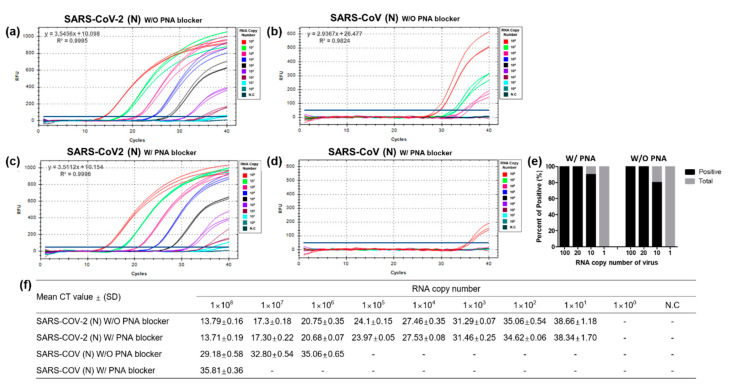
Optimization and evaluation of the peptide nucleic acid (PNA)-mediated RT-qPCR assay for N gene-specific amplification. Ten-fold serial dilutions (1 × 10^0^ to 1 × 10^8^ per 3 μL) of RNA transcripts of SARS-CoV-2 and SARS-CoV N genes without (**a**,**b**) and with (**c**,**d**) PNA blocker were processed in triplicates for RT-qPCR optimization. Each dilution was labeled with different colors to identify differences in the amplification curve, as shown in the figure. (**e**) Limit of detection in 20 repetitions using diluted RNAs (100, 20, 10, and 1 RNA copy numbers per 3 μL). (**f**) RT-qPCR-positive amplification was determined from the mean cycle threshold value for each RNA dilution point. N.C: negative control; Ct: cycle threshold; SD: standard deviation; “-”: not determined.

**Figure 4 diagnostics-10-00775-f004:**
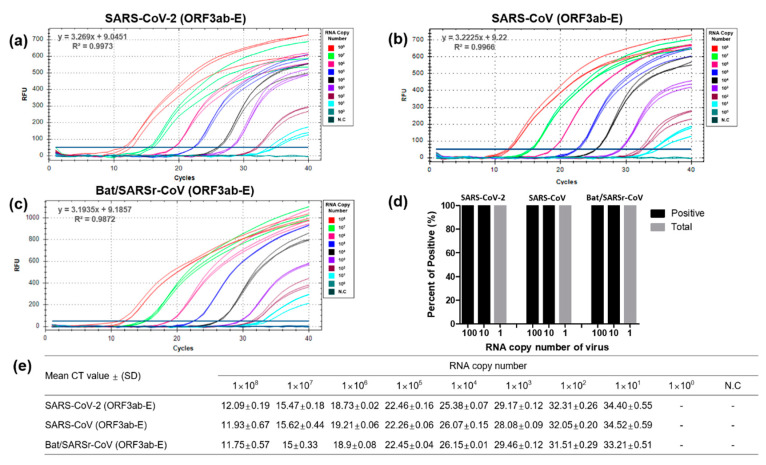
Optimization and sensitivity evaluation of the enhanced universal RT-qPCR assay to detect SARS-CoV-2 and SARS-CoV ORF3ab-E genes. Ten-fold serial dilution (1 × 10^0^ to 1 × 10^8^ per 3 μL) of RNA transcripts of SARS-CoV-2 (**a**), SARS-CoV (**b**), and Bat/SARSr-CoV (**c**) were processed in triplicates for RT-qPCR optimization. Each dilution was labeled with different colors to identify differences in the amplification curve, as shown in the figure. (**d**) Limit of detection in 20 repetitions using diluted RNAs (100, 20, 10, and 1 RNA copies per 3 μL). (**e**) RT-qPCR-positive amplification was determined from the mean cycle threshold value for each RNA dilution point. N.C: negative control; Ct: cycle threshold; SD: standard deviation; “-”: not determined.

**Figure 5 diagnostics-10-00775-f005:**
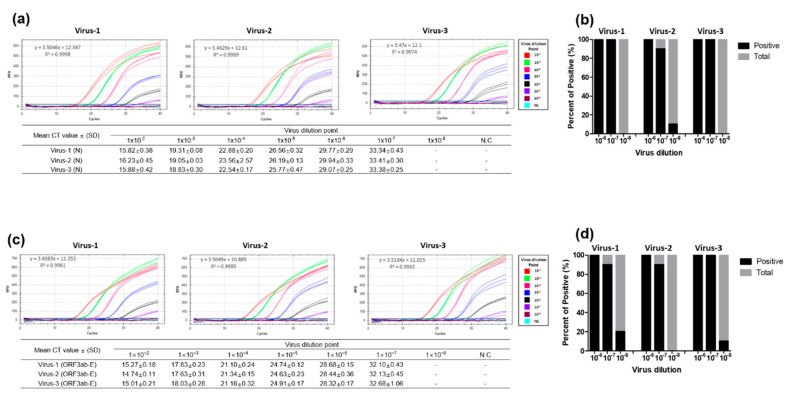
Sensitivity evaluation of the designed PNA-mediated RT-qPCR assay and the enhanced universal RT-qPCR method for detection using intact SARS-CoV-2 RNA. The evaluation was performed using RNA extracts from cell-propagated SARS-CoV-2 viruses isolated from patients diagnosed with COVID-19. Intact viral RNA extracts were ten-fold serially diluted (10^−2^ to 10^−8^) and processed for the detection of SARS-CoV-2 using the designed primers for N (**a**) and ORF3ab-E genes (**b**). RT-qPCR-positive amplification was determined from the mean cycle threshold value for each RNA dilution point. (**c**,**d**) Limit of detection was assessed using 10^−6^, 10^−7^, and 10^−8^ dilutions in ten repetitions. N.C: negative control; Ct: cycle threshold; SD: standard deviation; “-”: not determined.

**Table 1 diagnostics-10-00775-t001:** Oligonucleotide primers, probes, and blocker used in this study.

Assay Signature	Target Region	Primer Position	Primer/Probe	Sequence (5′ to 3′)	Concentration (Per Reaction)
PNA-mediated SARS-CoV-2 detection	Nucleocapsid	28371–28389 ^a^	Forward primer	GCGATCAAAACAACGTCGG	250 nM
	Nucleocapsid	28575–28593 ^a^	Reverse primer	ATACCATCTTGGACTGAGA	250 nM
	Nucleocapsid	28526–28545 ^a^	Probe	FAM-ACCGAAGAGCTACCAGACGA-BHQ	250 nM
	Nucleocapsid	28486–28502 ^a^	PNA blocker	*ATCAACACCAATAGTGG* ^c^	500 nM
		28342–28358 ^b^			
Universal SARSr-CoV detection	ORF3ab gene	26184–26201 ^a^	Forward primer	CCGACGACGACTACTAGC	250 nM
	Envelope protein	26365–26386 ^a^	Reverse primer	CTCACGTTAACAATATTGCAGC	250 nM
	Envelope protein	26329–26384 ^a^	Probe	FAM-TAGCCATCCTTACTGCGCTT-BHQ	250 nM

^a^ Values were based on the Beta-CoV/Korea/KCDC03/2020 sequence. ^b^ Values were based on the SARS-CoV/HKU-39849 sequence. ^c^ The italic characters indicate the PNA blocker sequence.

**Table 2 diagnostics-10-00775-t002:** Detection rate of the developed dual-target RT-qPCR for SARS-CoV-2, SARS-CoV, and Bat/SARSr-CoV.

Virus		Specimen Type	Number of Specimen	Detection Percentage (%)		Ct Value ^a^
				PNA-N Gene	ORF3ab-E Gene	
Coronavirus	SARS-CoV-2	Clinical	23 ^b^	82.6	100	18.17–37.93 ^c^
	MERS-CoV	Spike	1	ND	ND	15.9
	229E	Clinical/Spike	17	ND	ND	17.17–40.61
	NL63	Clinical/Spike	13	ND	ND	17.18–40.91
	OC43	Clinical/Spike	17	ND	ND	17.62–40.17
Influenza virus	Type B	Clinical/Spike	58	ND	ND	16.05–41.0
	H1N1	Clinical/Spike	17	ND	ND	15.84–41.15
	H3N2	Clinical/Spike	68	ND	ND	9.16–41.18
	HPAI H5NX	Spike	8	ND	ND	22.2–24.54
	H7N9	Spike	1	ND	ND	21.25
Other respiratory viruses	MPV	Clinical	11	ND	ND	19.10–39.49
	RSV A	Clinical	3	ND	ND	21.07–30.8
	RSV B	Clinical	46	ND	ND	16.77–41.97
	PIV	Clinical	8	ND	ND	27.93–39.75
	AdV	Clinical	3	ND	ND	27.96–40.21
	HRV	Clinical	37	ND	ND	15.00–40.00

^a^ The Ct values are based on the comparison of RT-qPCR detection of SARS-Cov-2 and other respiratory disease-causing viruses [22]. ^b^ Prior to the study, the 23 nasal swab samples were clinically diagnosed positive for SARS-CoV-2 by the Chungbuk National University. ^c^ Ct values were based on the detection range values from the 23 clinical samples using the PNA-N and ORF3ab-E primers. ND: not detected; SARS-CoV-2: severe acute respiratory syndrome-coronavirus-2; MERS-CoV: Middle East respiratory syndrome coronavirus; 229E: human coronavirus 229E; NL63: human coronavirus NL63; OC43: human coronavirus OC43; MPV: human metapneumovirus; RSVA: respiratory syncytial virus A; RSVB: respiratory syncytial virus B; PIV: parainfluenza virus; AdV: adenovirus; HRV: human rhinovirus.

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
