# Peer review of "Peptide Nucleic Acid (PNA)-Enhanced Specificity of a Dual-Target Real-Time Quantitative Polymerase Chain Reaction (RT-qPCR) Assay for the Detection and Differentiation of SARS-CoV-2 from Related Viruses"

_diagnostics, 2020, doi:10.3390/diagnostics10100775_

Round 1
Reviewer 1 Report
Won-Suk Choi et al. present a study in a timely topic in diagnostics. Nevertheless, despite the interest that the topic attracts, the presentation of results and their highlight can be improved for their interpretation. Overall, the presented method reduces the chance of false positives due to other Sarbecovirus in the clinical sample, even if such chance hasn't been reported as a major hindrance for diagnostics worldwide (a reference here could increase the perceived impact of the study); however, the method still allows for false positives as shown by their own results. Furthermore, it doesn't increase or improve the chances of detecting the true positives and false negatives, or discard the true negatives. Also, the current method uses a blocker that only binds to a specific sequence of SARS-CoV that could potentially accumulate mutations in the binding region (see below 1). In short, the novelty of the PNA blocker is marginal to the true purpose of the diagnostics method that depends on the performance of the RT-PCR. Aside from the PNA blocker, the performance of the designed primers should be contrasted against other available protocols (https://www.medrxiv.org/content/10.1101/2020.03.30.20048108v1.full.pdf).
Following, my suggestions about the text point by point:
1. The difference between SARS-CoV and SARS-CoV-2 in the target region of the PNA blocker depends on three mutations, two C->T and one G->A. Based on the accumulated data on Coronaviruses, the two C->U are very likely to happen (see https://www.biorxiv.org/content/10.1101/2020.05.01.072330v1), therefore removing the added value of the PNA blocker of the protocol.
2. In the abstract, does SARS-related coronaviruses refers to Sarbeco? Why the use of a distinct acronym? Does SARSr-CoVs include other SARS-CoV-2 isolated from animals? Please clarify this in the text and abstract.
3. In the whole text RNA copies is used indiscriminately without including the context of the volume size. Usually LOD is presented in RNA copies per microlitre. Similarly for dilutions, lines 141-142 need also to be updated with the volume.
4. Introduction lines 73-75: "it may not efficiently discriminate between the newly emergent coronavirus and other closely related viruses (i.e., SARS-CoV and other SARSr-CoV), pressing the need to develop a highly specific and sensitive viral detection method"; providing the reasons for the lack of discrimination power is important to justify the study. A reference or a simple note on the nucleotide similarity of the novel virus against previously existing virus would provide the context for this statement. Also, it is important to justify why the currently available RT-PCR is not enough for this task, although the presented method is basically RT-PCR.
5. Introduction lines 103-105: "This approach would improve the detection specificity for the SARS-CoV-2 N gene because of the arrested amplification of non-SARS-CoV-2 N genes by the PNA blocker"; however, MERS samples are non-SARS-CoV-2 N genes, and they are not blocked by the PNA blocker, maybe they are not detected as false positives but that's because of the specificity of the primers and the probe.
6. Results lines 152-154: "As hypothesized, the PNA blocker efficiently enhanced the specificity of the primer pairs and probe for N gene detection and amplification...", the specificity of the RT-PCR using these primers and probe was "enhanced", but the specificity of the primer pair or the probe remains unaffected, as the PNA blocker doesn't improve the interactions of each primer and probe with the sequence. In other words, this sentence needs revision to assure there is not confusion on the meaning of the "specificity" of the primers (unique binding of primers to target sequence) or the specificity of the method (accuracy detecting true negatives).
7. Discussion lines 271-273: "This amount of viral RNA would not be normally available in clinical samples as reported [34-37] suggesting that our PNA-mediated RT-qPCR will hardly demonstrate false positives from clinical sample of other SARSr-CoV"; this is not necessarily true as suggested by the report on SARS-CoV viral load above 10^9/mL in Hong Kong samples (https://www.ncbi.nlm.nih.gov/pmc/articles/PMC3367618/ figure 4). These cases with high viral loads will risk the performance of the presented method.
Reviewer 2 Report
Choi et al. described a peptide nucleic acid PNA)-based RT-qPCR assay targeting SARS-CoV-2 N gene enhanced the specificity of SARS-CoV-2 detection by 100-fold as compared to conventional RT-PCR while not losing the sensitivity.
Major comments:
Why human coronavirus HKU1 which is one of the four common coronaviruses cause common cold was not evaluated for cross-reactions?
Where 92.3% on line 247 came from? In the table, it is 82.6%.
Line 157-159 about LoD determination, could use statistical analysis (Probit) to determine. Otherwise, need to have 19 out of 20 replicates on the concentration of 20 RNA copies to be positive in order to claim the LoD.
The manuscript does not demonstrate that this PNA-based RT-qPCR alone is better than the conventional RT-PCR without the help of the universal ORF3ab-E because it would have missed 17.4%. There was no data either about using this method with dual-target could detect the positive SARS-CoV-2 clinical samples which were claimed by the conventional method are actually false-positives. Then why authors claimed this method has enhanced specificity?
Author Response
"Please see the attachment

Round 2
Reviewer 1 Report
The authors have addressed my main concerns. In my opinion the presented approach can attract the interest of the readership as alternative to improve the specificity of different tests.
Author Response
We would like to extend our sincerest gratitude for reviewing and helping us improve our manuscript. Thank you very much.
Reviewer 2 Report
Please show data on the same set of clinical samples using the NIID-Japan RT-qPCR gave a 74% detection rate. Are the samples missed by PNA-based RT-qPCR also missed by NIID-Japan RT-qPCR? Where is the data to demonstrate that those 23 clinical samples are all true SARS-CoV-2 positive samples? It seems authors provided the Ct values for those 23 clinical samples listed in Table 1: 18.17-37.93 and the footnote says "The Ct values are based on the NIID-Japan RT-qPCR detection method [33]". That means all 23 clinical samples were detected to be positive by NIID-Japan RT-qPCR. Then why authors say "the same set of clinical samples using the NIID-Japan RT-qPCR gave a 74% detection rate"? Very confusing.
Line 289: Should 13 be 23?
Author Response
Dear Reviewer:
We would like to extend our sincerest gratitude for evaluating our manuscript thoroughly. Through your comments and suggestions, we were able to improve our manuscript. We sincerely hope that through this response we can be able to address each concern one by one.
Thank you very much.
Sincerely,
Min-Suk Song
Reviewer 2:
Comments and Suggestions for Authors
Please show data on the same set of clinical samples using the NIID-Japan RT-qPCR gave a 74% detection rate. Are the samples missed by PNA-based RT-qPCR also missed by NIID-Japan RT-qPCR? Where is the data to demonstrate that those 23 clinical samples are all true SARS-CoV-2 positive samples? It seems authors provided the Ct values for those 23 clinical samples listed in Table 1: 18.17-37.93 and the footnote says "The Ct values are based on the NIID-Japan RT-qPCR detection method [33]". That means all 23 clinical samples were detected to be positive by NIID-Japan RT-qPCR. Then why authors say "the same set of clinical samples using the NIID-Japan RT-qPCR gave a 74% detection rate"? Very confusing.
- We highly appreciate the comment of the reviewer about this. To clarify, the 23 clinical samples (nasal swabs) were all from patients clinically diagnosed with COVID-19 (SARS-CoV-2 positive) by the Chungbuk National University Hospital, prior to the conduct of this study. These samples were tested using our primers for PNA-N gene and ORF3ab-E gene and cross-checked using the NIID-Japan RT-qPCR detection method. The Ct values mentioned in the table 1, 18.17-37.93 were the Ct values detection range of the study primers mentioned, PNA-N gene (82.6%) and ORF3ab-E gene (100%). While using the NIID-Japan RT-qPCR detection method, results showed only 17 out of 23 (73.9%) samples positive for SARS-CoV-2. This data of cross validation results was also added in the supplementary file in supplementary figure 3c. And the Results line 250 were clarified as this reads: “These 23 hospital confirmed clinical samples were also cross-checked using the National Institute of Infectious Diseases (NIID) Japan RT-qPCR assay method. From the cross validation, results showed that 17 out of 23 (73.9%) clinical samples were detected positive for SARS-CoV-2 with Ct value range: 18.10-36.40 (Supplementary Figure 3c).”
- We stand corrected for the mistakes made in the foot note in table 1 causing the confusion and lack of coherence. Hence to clarify, we corrected and added the footnotes in the table 1 (Line 259): “bPrior to the study, the 23 nasal swab samples were clinically diagnosed positive for SARS-CoV-2 by the Chungbuk National University.” We also added another footnote for clarification: “cCt values were based on the detection range values from the 23 clinical samples using the PNA-N and ORF3ab-E primers.”
Line 289: Should 13 be 23?
- We are very grateful for the keenness of the reviewer, we have failed to correct this line, now the text reads as: “All 23 clinical SARS-CoV-2 samples were detected at a range of 18.17 to 37.93 Ct values with a detection rate of 100% and 82.6% for universal ORF3ab-E and N gene, respectively (Table 1).”
Round 3
Reviewer 2 Report
The manuscript is being approved.